# Transition from Nurses to Medicalized Elderly Caregivers: Comparison on Willingness between Traditional and Modern Regions in China

**DOI:** 10.3390/ijerph19105950

**Published:** 2022-05-13

**Authors:** Wenqing Gao, Shuailong Li, Zhuoyuan Chi, Fangfang Gong, Wenxi Tang

**Affiliations:** 1Department of Pharmacoeconomics, School of International Pharmaceutical Business, China Pharmaceutical University, Nanjing 211198, China; 3321041218@stu.cpu.edu.cn (W.G.); 3221041344@stu.cpu.edu.cn (S.L.); 3220040603@stu.cpu.edu.cn (Z.C.); 2Center for Pharmacoeconomics and Outcomes Research, China Pharmaceutical University, Nanjing 211198, China; 3Department of Hospital Group Office, Shenzhen Luohu Hospital Group, Shenzhen 518005, China

**Keywords:** nurses, medicalized elder caregivers, transition willingness, influencing factors, China

## Abstract

As China is transitioning to an aging society, the Chinese government has proposed an eldercare pattern, called medicalized elderly care, to help solve the rapid aging and health care problems together. However, the shortage of elderly caregivers is a critical issue, with deficiency both in quantity and quality. This study aims to survey nurses’ willingness to transition into medicalized elderly caregivers and compare it between modern and traditional regions. Nurses working in Guangdong (modern region) and Jilin (traditional region) were investigated using a self-administered questionnaire in October 2021. We analyzed the influencing factors through χ²-test, *t*-test a and binary logistic regression model and further explored the influence of region using propensity score matching (PSM). A total of 1227 nurses were included, with 726 (59.2%) of them showing willingness to transition. Nurses from traditional regions showed a significantly higher willingness to transition after PSM (*p* = 0.027). Other factors influencing nurses’ willingness were age, education, lived with older adults, participated in voluntary activities related to older adults, visited eldercare institutions, attitudes toward older adults, knowledge about older adults, hospice care attitudes and death attitudes. The willingness of nurses to transition was not high enough. To have more willing and skillful human resources for eldercare, we need a more “intimate society for older adults” in the first place.

## 1. Introduction

The Seventh National Population Census Bulletin (No. 5), released by the National Bureau of Statistics of China, showed that 260 million Chinese people were aged 60 years or older in 2020, accounting for 18.70% of the total population [1]. The constant growth of the older adult population poses a challenge for eldercare resources in China [2]. Therefore, the Chinese government proposed an eldercare pattern that organically integrated pension and medical resources and closely linked up their service functions, called “medicalized elderly care (MEC)”, and actively provided a series of policy orientations. In 2021, the State Council of China issued “Opinions on Strengthening the Work on Aging in the New Era”, which noted that China should further advance the MEC and strengthen personnel team building [3]. There have been rapid developments in MEC in China. By the end of 2020, China had 5857 institutions (1.585 million beds) that can provide MEC services [4]. More than 90% of these eldercare institutions can provide various medical care services for registered older adults. However, many problems remain to be addressed. The quality of caregivers for the elderly in China was at a low level, as most of them were without sufficient knowledge and training on elderly care [5]. The shortage of workforce in terms of both quality and quantity is a prominent problem associated with MEC [6].

In response to strengthening personnel team building in eldercare, nursing personnel training systems in several developed countries have been gradually completed and could be used as references for China. Japan enacted a “Certified Social Worker and Certified Care Worker Act” in 1987, in which “certified care worker” was proposed as an occupation. Furthermore, Japan increased nursing skills operational training and clinical placements while incorporating home care, geriatric care and integrated care practices into nursing practice learning [7]. The multi-disciplinary training system has made Japanese care workers increasingly diverse, laid a solid foundation for elderly caregivers talent training and promoted the improvement of talent quality [8,9]. As for Denmark, schools involved nursing, natural sciences, health sciences, social sciences and human sciences in terms of discipline and professional settings. Students are able to acquire not only a broad theoretical foundation of physical therapy, but also nursing, management, religious and humanistic knowledge, to become compound nursing care talents [10]. Caregivers in Germany are called “care assistants”, and are required to learn psychological, legal, medical professional knowledge and elderly living room design. At the same time, many powerful elderly care enterprises have set up their own training schools, which not only train their own employees but also accept social members to join in. The diversified training methods have sought out potential nursing personnel and strengthened the eldercare staff [11]. Therefore, medicalized elderly caregivers (MEC givers) should be proficient in both professional nursing theory and skills. Moreover, they need learn comprehensive knowledge about older people and respect them. They must also safeguard older adults’ disease treatment and quality of life based on comprehensive services and professionalism, adopt a humanized nursing philosophy and provide hospice care to severely ill older adults [12]. In existing eldercare personnel teams in China, nurses from hospitals are the population best matched to the above-mentioned characteristics and the curricula they took; the work experience they had basically aligns with MEC requirements. They are expected to transition into caregivers and no additional education or training required, in order to strengthen the personnel team building in MEC as quickly as possible.

As of 2019, fewer than 100,000 geriatric nursing staff in China had obtained relevant qualification certificates [13]. Due to the expansion of the older population, the number of disabled people is continuously increasing. It was estimated that 40 million older people had difficulties in basic abilities of daily life in 2015 and this is expected to exceed 100 million by 2050 [14]. According to the internationally recognized standard (one member of nursing staff for every three older adults with disabilities), China still requires more than 13 million eldercare staff [15], which leaves a serious supply and demand imbalance. Therefore, the transition of nurses into MEC givers in China has become a necessary measure to solve the problem of an aging population. It is significant to explore the willingness of nurses to transition into MEC givers and find out the influencing factors to further improve their willingness.

Previously, Dai et al. [16] explored the willingness of 384 senior nurses from 15 hospitals to engage in long-term care for older adults in Henan Province, China, and analyzed their attitudes toward older adults and knowledge of them as influencing factors. In addition, Jang et al. [17] investigated 437 undergraduate nursing students’ willingness to care for older adults in South Korea and the United States, and analyzed influencing factors, such as the frequency of contacting with, sympathy for, and attitudes toward older adults. Furthermore, Che et al. [18] surveyed a total of 1462 nursing students from eleven nursing education institutions in Malaysia, and analyzed the relationship between their intention to work with older adults and their geriatric knowledge. However, most of the above-mentioned studies were conducted with nursing students; few studies have investigated whether nurses already working in hospitals were willing to transition into MEC givers. In addition, when exploring associated influencing factors, previous studies only considered attitudes toward or knowledge about older adults. There is a paucity of studies that comprehensively analyzed multiple factors, such as attitudes toward hospice care, experience of visiting eldercare institutions, participating in older adult-related voluntary activities and understanding of the pattern of MEC. Moreover, a multi-country study evaluated the willingness of nursing students from nine countries (or regions) to care for older adults and concluded that location was a more influential factor [19]. Due to the vast territory and economic disequilibrium in China, the east-to-west and north-to-south variations in the older population are obvious [20]. However, few studies have compared the willingness of nurses in different regions of China to transition into MEC givers. Therefore, our study aims to compare nurses’ willingness to transition into MEC givers in two provinces in China (Guangdong and Jilin) and analyzed influencing factors based on multiple perspectives. The results of the study will be helpful in strengthening personnel team building for MEC.

## 2. Methods

### 2.1. Study Design

In October 2021, a cross-sectional survey was conducted in two provinces (Guangdong and Jilin) in China. In total, 1400 nurses from three hospitals (two in Guangdong: Shenzhen Luohu People’s Hospital, Shenzhen Luohu Medical and Nursing Integrated Geriatrics Hospital; one in Jilin: Jilin Province People’s Hospital) were surveyed using self-designed questionnaires on their willingness of transitioning into MEC givers, attitude toward older adults, knowledge about older adults, hospice care attitude and death attitude.

### 2.2. Participants

#### 2.2.1. Sample Setting

In the first stage, from the perspective of location, we randomly selected Guangdong Province and Jilin Province to represent southern and northern regions. In the second stage, from the perspective of aging level, Shenzhen city in Guangdong and Changchun city in Jilin were selected to represent “modern” and “traditional” regions. Residents in Shenzhen aged 60 years or older in 2020 accounted for 5.36% of the total population which was much lower than the national level (18.7%), while that in Changchun accounted for 20.85% of the total population which was significantly higher than Shenzhen. In the third stage, considering the proportion of the number of third-class hospitals (22 in Shenzhen and 23 in Changchun), we randomly selected one hospital group in two cities (Shenzhen Luohu Medical and Nursing Integrated Geriatrics Hospital is actually a subsidiary hospital of Shenzhen Luohu People’s Hospital).

#### 2.2.2. Objectives

Participants were recruited using a convenient sampling and snowball sampling method. First, a sample of nurses from three hospitals were randomly selected as respondents and then they recommended other nurses for who the survey objectives were relevant to achieve a snowball effect. The inclusion criteria were nurses that: (1) were currently (at time of study) or previously engaged in geriatric nursing-related work; (2) gave informed consent and voluntarily participated in this questionnaire study; and (3) clearly expressed their own viewpoints and had good orientation to time, place and person. Participants that had a history of mental illness or disturbance of consciousness were excluded. A quality control question, a digital questionnaire platform and the time taken to answer the questionnaire were used to exclude those that were not answered seriously. Finally, 1400 nurses’ information was collected. Questionnaire Star software 2.0.91 (Survey Star Corp, Changsha, China) was used to import data from the collected electronic versions of the questionnaires into Excel format.

### 2.3. Instruments

The self-designed questionnaire used in this study comprised six parts. (1) General information, containing 13 items reflecting demographic characteristics (region, gender, age, religion, education, household registration, single child or not), whether they lived with older people, whether they had close contact with older adult relatives, and their knowledge of and attitudes toward MEC. (2) Willingness of transitioning from nurses to MEC givers. (3) Attitude toward older adults, assessed using Kogan’s Attitude toward Older People Scale (KAOPS). Developed by Kogan [21], the KAOPS contains 34 items and responses are on a 6-point Likert scale, with a total score of 34–204 points. A higher score indicates more positive attitudes toward older adults. (4) Knowledge about older adults, evaluated using the Facts on Aging Quiz (FAQ). Developed by Palmore [22], this FAQ scale contains 25 items and uses multiple options (“yes”, “no” and “do not know”) to determine participants’ knowledge, with a total score of 0–25 points. A higher score indicates better knowledge about older adults. (5) Hospice care attitude, assessed using the Frommelt Attitudes Toward Care of the Dying Scale Form B (FATCOD-B), which was developed by Frommelt [23]. The FATCOD-B contains 29 items, with responses on a 5-point Likert scale and a total scale score of 29–145. A higher score indicates more positive attitudes toward care of the dying. (6) Death attitude, evaluated using Death Attitude Profile Revised (DAP-R), compiled by Wong et al. [24]. This scale contains 32 items in five dimensions: fear of death (DF), death avoidance (DA), natural acceptance (NA), approach acceptance (AA) and escape acceptance (EA). Responses are on a 5-point Likert scale; a higher score for a dimension indicates the participant’s attitude is closer to that dimension. All the scales were translated into Chinese and were found to have excellent reliability and validity [25,26,27,28,29,30].

### 2.4. Statistical Analysis

#### 2.4.1. Descriptive Analysis

General data underwent descriptive statistical analysis, with count data expressed as *n* (%) and differences between two groups determined by χ²-test. Measurement data were expressed as means (M) ± standard deviation (SD), and between-group differences were evaluated using *t*-test.

#### 2.4.2. Regression Analysis

Using nurses’ willingness to transition into caregivers as the dependent variable, and general characteristics, the total KAOPS, FAQ, and FATCOD-B scores, and the scores for the five DAP-R dimensions as the independent variables, variables were entered into the regression model at α = 0.05 and excluded from the regression model at α = 0.10.

The basic form of the logistic regression model is given by Equation (1): (1)$$P_{i}=F\left(y\right)=\frac{e^{\alpha +\sum_{i=1}^{n}\beta_{i}x_{i}}}{1+e^{\alpha +\sum_{i=1}^{n}\beta_{i}x_{i}}}$$

Logistic transformation of Equation (1) gives a linear regression model between the probability function and independent variables, as shown in Equation (2): (2)$$\ln\frac{P_{i}}{1-P_{i}}=\alpha +\overset{n}{\underset{i=1}{\sum }}\beta_{i}x_{i}$$

$P_{i}$ represents the probability of nurses transitioning into MEC givers; y is the dependent variable, indicating whether nurses were willing to transition into MEC givers (willing = 1 and unwilling = 0); $x_{i}$ is the independent variable (the i-th influencing factor); and $\beta_{i}$ is the regression coefficient of the independent variable. All results were considered statistically significant at *p* < 0.05. SPSS version 22.0 (IBM Corp., Armonk, NY, USA) was used for the statistical analyses.

#### 2.4.3. Region Differences Comparisons

To compare the influence of traditional and modern regions on nurses’ willingness, we performed propensity score matching (PSM) to control for confounding factors. Using nearest-neighbor matching, participants were grouped by workplace and matched for all other factors, with a matching tolerance of 0.02.

## 3. Results

### 3.1. Basic Characteristics and Willingness to Transition

Table 1 shows the basic characteristics of 1227 included respondents. The overall rate of willingness to transition was 59.2% (*n* = 726). Factors with significant differences in nurses’ willingness to transition into MEC givers were age (*p* = 0.001), living with older people (*p* < 0.001), having close contact with older adult relatives (*p* < 0.001), participating in older-adult-related voluntary activities (*p* < 0.001), visiting eldercare institutions, such as nursing homes (*p* < 0.001), knowing the eldercare pattern of MEC (*p* < 0.001) and agreeing to implement the MEC in China (*p* = 0.031).

### 3.2. Comparison of Scale Scores by Willingness to Transition into Caregivers

Table 2 shows mean scores and standard deviations for two different groups of nurses who were willing/unwilling to transition into MEC givers for each of the scales. We found that higher scores of KAOPS, FAQ and FATCOD-B (all ***p***s < 0.001) were significantly associated with higher willingness.

### 3.3. Factors Influencing Nurses’ Transition Willingness

Table 3 shows factors that influenced nurses’ willingness for transitioning into MEC givers. We found that nurses from Jilin were more willing than those from Guangdong. Nurses between the ages of 31 and 39 had less willingness than ≥40. Nurses with the highest education in technical secondary schools or below were more willing than those who had masters’ degrees. Nurses who lived with older people, had close contact with older adult relatives, participated in voluntary activities related to older adults and visited eldercare institutions, such as nursing homes, had more willingness than those who had not experienced these. The higher scores of KAOPS, FAQ, FATCOD-B and DAP-R indicated that those nurses were more willing to transition into MEC givers.

### 3.4. Influence of Regional Factors on Nurses’ Willingness to Transition into Caregivers

To further explore the influence of nurses’ region, we balanced confounding factors using PSM and then examined differences in nurses’ willingness to transition into caregivers between Guangdong and Jilin, and the results are shown in Table 4.

Finally, 335 pairs (770 participants) were matched successfully among 1227 participants. Before matching, there were significant differences in gender, age, education, household registration, etc. (all ***p***s < 0.05). After matching, no significant differences were observed between the two groups in any analyzed factor (all ***p***s > 0.05), with the exceptions of age and education. After matching, nurses in Jilin (*n* = 218, 65.1%) still showed a higher willingness to transition compared to those in Guangdong (*n* = 190, 56.7%), indicating that region had a prominent influence on nurses’ willingness to transition into MEC givers.

## 4. Discussion

The results of this study provided insights into how hospital nurses were willing to transition into MEC givers in China. The main findings of this study are as follows. First, less than 60% of the nurses surveyed were willing to transition into MEC givers. Second, nurses’ willingness is highly associated with their attitude towards death and experiences with older people. Personal contact with older people allows nurses to gain deeper understanding of older adults, which may reverse incorrect stereotypes regarding the job of an elderly caregiver. Further, attitudes toward hospice care and death and knowledge of older people are essential factors that influence nurses’ decisions about working as professional caregivers. The above finding is consistent with many other studies [31,32,33,34,35] and can be explained by the “knowledge–attitude–behavior” theory [36]. The profound influence of traditional culture and Confucianism mean that Chinese people attach importance to the interdependence between generations. Nurses who were mainly cared for by older people when they were young had more positive attitudes toward older adults than nurses who were not cared for by older people [37]. Several studies indicated that knowledge about aging plays a major role in professionals’ positive attitudes [38,39,40,41]. Developing correct knowledge of aging is conducive for caregivers to cultivate optimistic attitudes toward older adults and hospice care, and these positive attitudes help improve people’s beliefs and behaviors.

Interestingly, our finding indicates that nurses from the traditional region (Jilin) had higher willingness to transition into MEC givers than nurses from a relatively modern region (Guangdong). The two regions in this study, which were selected from the northeastern and southern parts of China, have marked differences in their geographic location, customs, culture and economic development level. Guangdong on the southern coast is a province with a developed economy, large population and high urbanization level. The migrant population accounts for 17% of the total population in Guangdong, meaning it is regarded as a modern “stranger society”. Jilin, in the northeastern region, has a relatively low level of economic development, with most of the population being concentrated in rural areas, and it is considered a traditional “acquaintance society”. In the present study, we found a significant difference in nurses’ willingness to transition into caregivers between Guangdong and Jilin before PSM (*p* = 0.006). To verify the influence of region on nurses’ willingness, we used PSM to control for independent variables, such as demographic characteristics and associated life experiences, thereby reducing the interference of confounding factors on the regional difference in nurses’ willingness. Among the participants after PSM at 1:1, we observed a significantly higher proportion of nurses willing to transition into caregivers in Jilin than in Guangdong (65.1% vs. 56.7%; *p* = 0.027). Therefore, there was a considerable difference in nurses’ willingness to transition into caregivers in the northeastern and southern provinces of China based on regional characteristics.

While Mohammad et al. [42] found that most nurse participants have at least one misconception (88.6%) and hold at least one negative attitude (89.8%), which reveals that even in a country with high religious observance and close family ties, ageism exists in healthcare settings because of nurses’ poor knowledge and attitudes toward older adults. Dai et al. [16] surveyed willingness to engage in long-term care for older adults and influencing factors among 384 senior nurses. They found that more than 60% of nurse participants were unwilling to engage in long-term care for older adults, with this unwillingness mainly related to self-rated health, being busy taking care of family and salary. There were some differences between our results and those of previous studies. It can be inferred that the environmental change and salary adjustment faced by occupational transition may be more sensitive topics among young people. Therefore, we suggest that hospitals or communities may conduct lectures for MEC givers, and provide relevant publicity posters, so that nurses can learn about the caregiver occupation outside of work. Nurses who had participated in older-adult-related activities held more positive attitudes toward transitioning into the caregiver occupation. These close correlations suggested that effective human resource management of caregivers should comprehensively cover relevant knowledge, attitudes and behaviors. The differing transition willingness of nurses between Guangdong and Jilin indicates that the central and local governments of China should conduct comprehensive planning in the context of vast national territory and also adjust measures to local conditions based on localized characteristics when designing and planning personnel team building of caregivers to integrate medical care and eldercare services. However, there are many other influencing factors that may underlie the regional difference (e.g., regional economic development level, traditional customs, and nurses’ salary), which need to be explored in further research.

To our knowledge, few studies have used PSM as a statistical method to explore the relationship between geographical region and nurses’ willingness to transition into MEC givers. The present study compared the differences in nurses with transition willingness between traditional and modern regions innovatively, and balanced confounding factors. This study also combined a multi-dimensional self-designed questionnaire with the KAOPS, FAQ, FATCOD-B and DAP-R scales to investigate the influencing factors, as well as the possible measurements to help improve willingness based on large sample data. However, this study also had some limitations. First, this is a cross-sectional design, which is a one-time data collection study that failed to determine the causality relationship between variables, so further research should use a longitudinal study, at various time points, to understand the dynamics in willingness to transition. Second, the use of convenience sampling and random sampling may also affect the likelihood of selection bias. The respective samples for each city were only chosen in one or two hospitals, which cannot represent the entire population of nurses from hospitals in that region. Third, the self-designed scale did not fully survey all possible factors influencing nurses’ willingness, such as salary level and job department.

## 5. Conclusions

This study explored Chinese nurses’ willingness to transition into MEC givers and analyzed factors influencing willingness to transition based on multiple perspectives. The outcomes of the study may assist personnel identification to support the pattern of MEC in China during aging transformation. That is, the main force of MEC going forward will be nurses that have experience of intimate contact with older people, sufficient knowledge of and positive attitudes toward older adults, aging, hospice care, and death, and that live in an “acquaintance society”, characterized by “human culture”. As the age of nurses increases, their willingness to transition into caregivers also increases. Therefore, we suggest that nurses should actively participate in older-adult-related practical activities and cultivate a positive service awareness during close contact with older people. In addition, hospitals and communities should implement publicity activities, such as posters or lectures related to MEC givers. Furthermore, subsequent research should consider evaluating the adequacy of older-adult-related practical activities in providing older-adult-care knowledge and skills and their implementation plans to identify training gaps.

## Figures and Tables

**Table 1 ijerph-19-05950-t001:** Participants’ general information and willingness to transition into MEC givers.

Variables	Cases (%)	Willingness (%)	χ²	*p*
Region			0.015	0.903
Guangdong Province	693 (56.5)	409 (59.0)		
Jilin Province	534 (43.5)	317 (59.4)		
Gender			0.100	0.752
Male	56 (4.6)	32 (57.1)		
Female	1171 (95.4)	694 (59.3)		
Age			13.127	**0.001**
≤30	388 (31.6)	228 (58.8)		
31~39	514 (41.9)	280 (54.5)		
≥40	325 (26.5)	218 (67.1)		
Education			3.187	0.364
Technical secondary school or below	37 (3.0)	27 (73.0)		
Junior college	286 (23.3)	170 (59.4)		
Bachelor’s degree	877 (71.5)	514 (58.6)		
Master’s degree or above	27 (2.2)	15 (55.6)		
Household registration			0.098	0.755
Urban areas	591 (48.2)	347 (58.7)		
Rural areas	636 (51.8)	379 (59.6)		
Religious belief			0.023	0.879
Yes	117 (9.5)	70 (59.8)		
No	1110 (90.5)	656 (59.1)		
Only child or not			0.031	0.859
Yes	320 (26.1)	188 (58.8)		
No	907 (73.9)	538 (59.3)		
Whether they lived with older people			17.525	**<0.001**
Yes	955 (77.8)	595 (62.3)		
No	272 (22.2)	131 (48.2)		
Whether they had close contact with older adult relatives			42.861	**<0.001**
Yes	959 (78.2)	614 (64.0)		
No	268 (21.8)	112 (41.8)		
Whether they participated in older adult-related voluntary activities			75.246	**<0.001**
Yes	498 (40.6)	368 (73.9)		
No	729 (59.4)	358 (49.1)		
Whether they visited eldercare institutions such as nursing homes			53.745	**<0.001**
Yes	578 (47.1)	405 (70.1)		
No	649 (52.9)	321 (49.5)		
Whether they knew the eldercare pattern of MEC			15.904	**<0.001**
Yes	902 (73.5)	564 (62.5)		
No	325 (26.5)	162 (49.8)		
Whether they agreed to implement the MEC in China			4.670	**0.031**
Yes	998 (81.3)	605 (60.6)		
No	229 (18.7)	121 (52.8)		

**Table 2 ijerph-19-05950-t002:** Comparison of attitude scale scores by willingness/unwillingness to transition into MEC givers.

Variables	Willingness (*n* = 726)	Unwillingness (*n* = 501)	t	*p*
KAOPS score	133.97 ± 18.34	123.36 ± 17.19	10.218	**<0.001**
FAQ score	11.69 ± 3.10	10.58 ± 3.37	5.918	**<0.001**
FATCOD-B score	97.90 ± 10.18	92.43 ± 7.34	10.321	**<0.001**
DAP-R scale				
Death fear DF	20.10 ± 5.13	20.67 ± 4.91	−1.963	0.05
Death avoidance DA	15.42 ± 3.89	15.56 ± 3.84	−0.657	0.512
Natural acceptance NA	18.18 ± 3.47	17.98 ± 3.45	0.984	0.325
Approach acceptance AA	29.89 ± 7.02	29.44 ± 6.75	1.138	0.255
Escape acceptance EA	14.24 ± 4.08	14.65 ± 3.95	−1.746	0.081

**Table 3 ijerph-19-05950-t003:** Binary logistic regression analysis of nurses’ willingness to transition.

Variables	B	SE	Wald χ²	*p*
Region (Guangdong)				
Jilin	0.444	0.332	0.637	**0.006**
Age (≥40)				
≤30	−0.372	0.197	3.549	0.060
31~39	−0.409	0.176	5.395	**0.020**
Education (Master’s degree or above)				
Technical secondary school or below	1.617	0.634	6.514	**0.011**
Junior college	0.892	0.503	3.147	0.076
Bachelor’s degree	0.756	0.487	2.406	0.121
Whether they lived with older people (No)				
Yes	0.476	0.170	7.833	**0.005**
Whether they had close contact with older adult relatives (No)				
Yes	0.434	0.167	6.797	**0.009**
Whether they participated in voluntary activities related to older adults (No)				
Yes	0.737	0.155	22.672	**<0.001**
Whether they visited eldercare institutions such as nursing homes (No)				
Yes	0.408	0.148	7.602	**0.006**
KAOPS score	0.030	0.004	46.420	**<0.001**
FAQ score	0.059	0.022	7.430	**0.006**
FATCOD-B score	0.064	0.010	45.197	**<0.001**
DAP-R scale				
Natural acceptance NA	0.081	0.024	11.015	**0.001**
Approach acceptance AA	0.037	0.016	5.343	**0.021**

**Table 4 ijerph-19-05950-t004:** Comparison of willingness to transition in Guangdong and Jilin before and after matching.

Variables	Before PSM		*p*	After PSM		*p*
	Guangdong (*n* = 693)	Jilin (*n* = 534)		Guangdong (*n* = 335)	Jilin (*n* = 335)	
Whether they had willingness to transition into caregivers						
Yes	409	317	0.006	190	218	**0.027**
Gender			<0.001			0.247
Male	45	11		17	11	
Female	648	523		318	324	
Age			<0.001			**0.004**
≤30	300	88		87	71	
31~39	239	275		131	174	
≥40	154	171		117	90	
Education			<0.001			**0.008**
Technical secondary school or below	29	8		12	4	
Junior college	182	104		64	73	
Bachelor’s degree	460	417		245	255	
Master’s degree or above	22	5		14	3	
Household registration			<0.001			0.487
Urban areas	250	341		164	173	
Rural areas	443	193		171	162	
Only child or not			<0.001			1.000
Yes	89	231		82	82	
No	604	303		253	253	
Whether they lived with older people			<0.001			0.644
Yes	577	378		262	257	
No	116	156		73	78	
Whether they participated in voluntary activities related to older adults			<0.001			0.752
Yes	314	184		135	131	
No	379	350		200	204	
Whether they visited eldercare institutions such as nursing homes			<0.001			0.877
Yes	366	212		159	157	
No	327	322		176	178	
Whether they knew the eldercare pattern of MEC			<0.001			0.605
Yes	539	363		239	245	
No	154	171		96	90	
KAOPS scale	131.32 ± 17.70	127.45 ± 19.55	<0.001	128.80 ± 17.05	128.32 ± 20.56	0.744
FATCOD-B scale	96.66 ± 9.56	94.38 ± 9.31	<0.001	95.34 ± 9.37	95.52 ± 9.78	0.812
DAP-R scale						
Escape acceptance EA	14.16 ± 3.92	14.72 ± 4.15	**0.017**	14.66 ± 3.96	14.66 ± 4.31	0.985

## Data Availability

All data that support the findings of this study are available from the corresponding author upon reasonable request.

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
