# Peer review of "Transition from Nurses to Medicalized Elderly Caregivers: Comparison on Willingness between Traditional and Modern Regions in China"

_ijerph, 2022, doi:10.3390/ijerph19105950_

Round 1

Reviewer 1 Report

I thank the authors for giving me the opportunity to read their manuscript.

The topic is of interest to the magazine. However, the manuscript is not well written in some parts, although it does contain a message about an interesting aspect of Chinese caregivers.

The article presented will have to be revised in some of its parts.

As far as the introduction is concerned, the authors should make considerations at an international and not just a local level, also inserting international reference literature.

In addition, authors should better indicate the purpose of the study.

In the discussion the authors should make further comparisons with other similar international studies, comparing their data with international ones.

Bibliographic references should be changed because as the reference literature concerns many Chinese articles, in my opinion they should also include reference bibliography with a wider international field, given that the journal's readers target is international.

The final consideration, I believe that with the proposed changes, the document is an acceptable case for publication.

Best regards

Author Response

Dear reviewer,

We sincerely thank you for your careful reading and examination, your professional suggestions are of great help in improving the quality of our study (Transition from nurses to medicalized elderly caregivers: comparison on willingness between traditional and modern regions in China).

We have carefully considered all professional comments from you and revised our manuscript accordingly. The manuscript has also been double-checked, and the typos and grammar errors we found have been corrected. In the following section, we summarize our responses to each comment from your reviewers. We believe that our responses have well addressed all concerns from the reviewers. We hope our revised manuscript can be accepted for publication.

Following are your advice and my modification:

Point 1: As far as the introduction is concerned, the authors should make considerations at an international and not just a local level, also inserting international reference literature.

Response 1: We have added some international references when described the cultivation of nursing caregivers in other countries in the second paragraph of the Introduction (Reference 8-11) and added more international studies of nursing students’ willingness to care for older adults in the fourth paragraph of the Introduction (Reference 18,19).

Point 2: In addition, authors should better indicate the purpose of the study.

Response 2: We have further explained the purpose of the study in the last paragraph of the introduction, “Therefore, our study aims to compare nurses’ willingness to transition into MEC givers in two provinces in China: Guangdong, Jilin and analyzed influencing factors based on multiple perspectives.” and indicated the possible reason for the differences in willingness brought by regions.

Point 3: In the discussion the authors should make further comparisons with other similar international studies, comparing their data with international ones.

Response 3: We have added some more international and more latest studies as comparisons for qualitative research in Discussion section (Reference 31-35), and added a study as a comparison for quantitative research (Reference 42).

Point 4: Bibliographic references should be changed because as the reference literature concerns many Chinese articles, in my opinion they should also include reference bibliography with a wider international field, given that the journal's readers target is international.

Response 4: We have changed most of Chinese bibliographic references and replaced them with international journals or government documents as much as possible (Reference 2,5,6,9-11,18-20,31-35,42).

Reviewer 2 Report

Thank you for the opportunity to review this paper. Overall, the paper is clear and well-written. The methods are a clear fit for the research question and robust. Results are presented well. The discussion and conclusions discuss both the contribution of the study to the science and possible policy recommendations.

My major concern is in the introduction. While it is clear that such a study has never been conducted before, I am unclear as to why this study needs to be conducted. In other words, why did the authors think that a difference in willingness to transition to MECs would differ by region? Why is it significant? Please see detailed comments below:

Abstract:

Well-written abstract

Introduction:

General: The authors provide a thorough explanation of MECs and its policy significance.

Line 87: Is “transit” the correct word? I think you may mean “transition.” If so, this change will need to be made throughout.

Line 85-89: Would nurses transitioning from a hospital to an MEC setting require additional education or training?

Line 96: Please clarify what is meant by “modern province” and “traditional province.” While I know that this is discussed in the methods section, it would be helpful to the reader to know what is intended at the first mention.

General: An explanation of the motivation to study regional differences in willingness to transition to MECs would strengthen this introduction considerably.

Methods:

Robust and fit the research question

Section 2.2.2: With snowball sampling, how did the researchers reach the 1400 nurses?

Section 2.3: Was the “self-designed” questionnaire piloted among a subset of participants before survey dissemination to all participants? If so, did this affect the survey content?

Results:

Well-presented and tables are comprehensive

Discussion:

Conclusions fit well with the results. The discussion covers both contributions to the current body of literature and policy recommendations.

Conclusion:

The content is good, though I’d comment that the conclusion is a bit long

Author Response

Dear reviewer,

We sincerely thank you for your careful reading and examination, your professional suggestions are of great help in improving the quality of our study (Transition from nurses to medicalized elderly caregivers: comparison on willingness between traditional and modern regions in China).

We have carefully considered all professional comments from you and revised our manuscript accordingly. The manuscript has also been double-checked, and the typos and grammar errors we found have been corrected. In the following section, we summarize our responses to each comment from your reviewers. We believe that our responses have well addressed all concerns from the reviewers. We hope our revised manuscript can be accepted for publication.

Following are your advice and my modification:

Point 1: My major concern is in the introduction. While it is clear that such a study has never been conducted before, I am unclear as to why this study needs to be conducted. In other words, why did the authors think that a difference in willingness to transition to MECs would differ by region?

Response 1: Thanks for your professional question. We have further explained why our study needs to be conducted in the last paragraph of the Introduction section of the revised manuscript (Line 100-108). At the study design stage, we read a multi-country study on the willingness of nursing students to care for older people and the study concluded location was a influential factor, we would like to explore whether location is still a significant influential factor in China. Therefore, we put “region” as a independent variable into regression model, further used propensity score matching to balance other variables to only explore the influence of region.

Point 2: Line 87: Is “transit” the correct word? I think you may mean “transition.” If so, this change will need to be made throughout.

Response 2: We are sorry for the incorrect statement. We have modified “transit” to “transition” and modified its other forms accordingly in the full text.

Point 3: Would nurses transitioning from a hospital to an MEC setting require additional education or training?

Response 3: Thanks for your professional question. We have introduced the education and training that MEC givers need in the second paragraph of the Introduction section. It is partially aligns with the curriculum that nurses take in their education. In existing eldercare personnel teams in China, we think nurses are the population best matched to the characteristics that MEC givers require. Therefore, we explore the willingness of nurses from hospitals to transition into MEC gives. As for your question, our study was conducted under the assumption that no additional education or training is required. Therefore, we think it is not necessary to discuss whether nurses transition from a hospital to an MEC setting require additional education or training.

Point 4: Please clarify what is meant by “modern province” and “traditional province.” While I know that this is discussed in the methods section, it would be helpful to the reader to know what is intended at the first mention.

Response 4: Thanks a lot for your meaningful suggestion. We have deleted the statement of “modern” and “traditional” in the Introduction section, and further explained its meaning in the Methods section (2.2.2. Sampling Setting).

Point 5: An explanation of the motivation to study regional differences in willingness to transition to MECs would strengthen this introduction considerably.

Response 5: We have added the reason we conduct this study in the last paragraph of the Introduction section and further explained the choice of two regions in the Method section (2.2.2. Sample Setting).

Point 6: Section 2.2.2: With snowball sampling, how did the researchers reach the 1400 nurses?

Response 6: Thanks for your professional question. We have added sampling method in 2.2.2. Objectives: “Participants were recruited using a convenient sampling and snowball sampling method. First, a part of nurses from three hospitals were randomly selected as respond-ents and then they recommended other nurses who belonged to the survey objectives to achieve a snowball effect.”.

Point 7: Section 2.3: Was the “self-designed” questionnaire piloted among a subset of participants before survey dissemination to all participants? If so, did this affect the survey content?

Response 7: Thanks for your professional question, but we didn’t do this.

Reviewer 3 Report

Introduction: This part is correct, some references too older.

Study Design: In this section it would be missing to describe the type of study, at least to say that it is a descriptive cross-sectional study. Explain which of the hospitals are from Guangdong and which from Jilin.

Sample Setting:in this part does not describe the justification for the choice o:f these 3 hospitals as in the participants, nor does it describe how many hospitals there are in these two cities.

Sampling: The selection method, which due to the size achieved could have been random sampling, avoiding the selection bias present in this study and which the authors have not identified as a limitation either.

Quality Control:In quality control, data protection regulations, for example, must be taken into account. 

Cluster analysis could have been done if we knew the participants from each hospital, which are not described in the sample and may also be a confounding variable.

The main problem I see, and the authors do not describe it, is in the selection of the sample, they do not explain why there are three hospitals, they do not explain why these cities, we do not know the nursing population in the different regions or the hospitals there are to be able to make inferences, and they do not recognise this as a limitation either.

Author Response

Dear reviewer,

We sincerely thank you for your careful reading and examination, your professional suggestions are of great help in improving the quality of our study (Transition from nurses to medicalized elderly caregivers: comparison on willingness between traditional and modern regions in China).

We have carefully considered all professional comments from you and revised our manuscript accordingly. The manuscript has also been double-checked, and the typos and grammar errors we found have been corrected. In the following section, we summarize our responses to each comment from your reviewers. We believe that our responses have well addressed all concerns from the reviewers. We hope our revised manuscript can be accepted for publication.

Following are your advice and my modification:

Point 1: Introduction: This part is correct, some references too older.

Response 1: Thanks for your professional suggestion. We have changed some old references to more latest and international references in the revised manuscript (Reference 2,5,6,10,11).

Point 2: Study Design: In this section it would be missing to describe the type of study, at least to say that it is a descriptive cross-sectional study. Explain which of the hospitals are from Guangdong and which from Jilin.

Response 2: Thanks for your professional suggestion. We have added the description of the type of study in the Methods section (2.1. Study Design) and further illustrated the information of three hospitals.

Point 3: Sample Setting:in this part does not describe the justification for the choice of these 3 hospitals as in the participants, nor does it describe how many hospitals there are in these two cities.

Response 3: Thanks for your professional suggestion. We have added more information about the justification for the choice of these 3 hosipital and described the number of hospitals in these two cities respectively (2.2.2. Sample Setting, Line 127-131)

Point 4: Sampling: The selection method, which due to the size achieved could have been random sampling, avoiding the selection bias present in this study and which the authors have not identified as a limitation either.

Response 4: Thanks for your professional question.We are sorry that we have ignored this problem in the original manuscript. We have added this as a limitation in the last paragraph of the Discussion (Line 308-311).

Point 5: Quality Control:In quality control, data protection regulations, for example, must be taken into account.

Response 5: We are sorry that we can’t understand your suggestion well. If you were talking about the privacy protection of the respondents, we have covered it in the self-designed questionnaire.

Round 2

Reviewer 2 Report

Thank you for submitting this revised manuscript. I believe that my comments have been addressed sufficiently. The motivation for the study is much clearer. I will reiterate that I believe that the study design is robust. These results indeed provide insight into factors associated with decisions to transition into medicalized elderly care. Though, I believe that these factors may be interesting for a global audience as this information may be interesting to stakeholders interested in attracting more nurses into care settings for older adults.

I have one more minor comment regarding this new draft:

I appreciate the additions in the background regarding educational requirements/training. I think it would be helpful to add a clear statement that nurses in China are not required to attain additional education/training/certification to work in MECs.

Once again, thank you for the opportunity to review this manuscript

Author Response

Dear reviewer,

We sincerely thank you for your careful reading and examination, your professional suggestions are of great help in improving the quality of our study (Transition from nurses to medicalized elderly caregivers: comparison on willingness between traditional and modern regions in China).

We have carefully considered all professional comments from you and revised our manuscript accordingly. The manuscript has also been double-checked, and the typos and grammar errors we found have been corrected. In the following section, we summarize our responses to each comment from your reviewers. We believe that our responses have well addressed all concerns from the reviewers. We hope our revised manuscript can be accepted for publication.

Following are your advice and my modification:

Point 1: I think it would be helpful to add a clear statement that nurses in China are not required to attain additional education/training/certification to work in MECs.

Response 1: Thanks for your professional suggestion. We have added a clear statement in the end of the second paragraph of the Introduction section (Line 72-76): “Nurses from hospitals are expected to transition into caregivers and no additional education or training required, so that can strengthen the personnel team building in MEC as quickly as possible.”
